# Poloxamer Hydrogels for Biomedical Applications

**DOI:** 10.3390/pharmaceutics11120671

**Published:** 2019-12-10

**Authors:** Eleonora Russo, Carla Villa

**Affiliations:** Department of Pharmacy, University of Genoa, Viale Benedetto XV, 16132 Genova, Italy; villa@difar.unige.it

**Keywords:** poloxamer, hydrogels, micelle, thermosensitive, biomedical, copolymer

## Abstract

This review article focuses on thermoresponsive hydrogels consisting of poloxamers which are of high interest for biomedical application especially in drug delivery for ophthalmic, injectable, transdermal, and vaginal administration. These hydrogels remain fluid at room temperature but become more viscous gel once they are exposed to body temperature. In this way, the gelling system remains at the topical level for a long time and the drug release is controlled and prolonged. Poloxamers are synthetic triblock copolymers of poly(ethylene oxide)-b-poly(propylene oxide)-b-poly(ethylene oxide) (PEO-PPO-PEO), also commercially known as Pluronics^®^, Synperonics^®^ or Lutrol^®^. The different poloxamers cover a range of liquids, pastes, and solids, with molecular weights and ethylene oxide–propylene oxide weight ratios varying from 1100 to 14,000 and 1:9 to 8:2, respectively. Concentrated aqueous solutions of poloxamers form thermoreversible gels. In recent years this type of gel has arouse interest for tissue engineering. Finally, the use of poloxamers as biosurfactants is evaluated since they are able to form micelles in an aqueous environment above a concentration threshold known as critical micelle concentration (CMC). This property is exploited for drug delivery and different therapeutic applications.

## 1. Introduction

The word “hydrogel”, according to Lee, Kwon and Park is due to an article published in 1894, but the first crosslinked network material that appeared in literature and that has been described by its typical hydrogel properties, was a polyhydroxyethylmethacrylate (HEMA) hydrogel developed much later, in 1960, by O. Wichterle and D. Lim, with the aim of using it in permanent contact applications with human tissues, i.e., as soft contact lenses [1].

Since then, hydrogels have been used as systems for drug controlled delivery, to facilitate the localized, sustained and prolonged release of a drug, thereby decreasing the number of administrations, avoiding side effects and following low doses [2].

The most widely studied environmentally responsive systems were temperature sensitive hydrogels, in which physical entanglements, hydrogen bonding, and hydrophobic interactions are the main features that constitute the crosslinks. Two different types of thermo-sensitive hydrogels exist that undergo gelation either by cooling below the upper critical gelation temperature (UCGT) or by heating above the lower critical gelation temperature (LCGT), respectively. Hydrogels with LCGT behavior and sol-to-gel transition at 37 °C have gained increasing attention in the biomedical field as carriers for cells, drugs, and biomolecules, since they allow encapsulation in mild conditions (temperature ≤ 37°C) [3]. 

Poloxamers and poloxamines are examples of these LCGT biocompatible thermoreversible hydrogels that were introduced in the 1950s by BASF (Iselin, NJ, USA) when they started being used for detergent development, but also in other areas, like agriculture, food, and paints [4].

Poloxamers or Pluronics^®^ are a class of water-soluble nonionic triblock copolymers formed by polar (poly ethylene oxide) and non-polar (poly propylene oxide) blocks. which confer amphiphilic and surface active properties to the polymers. Their aqueous solutions undergo sol-to-gel transition with increasing the temperature above a LCGT; moreover, the coexistence of hydrophilic and hydrophobic monomers into block copolymers allows the formation of ordered structures in solution, the most common of these being the micelles. The formation of micelles in solution is a reversible and dynamic process useful for encapsulating hydrophobic drugs and delivering them into an aqueous environment.

They can be considered as smart polymers, for their stimuli-sensitive properties, due to the different behaviors of these polymers since they can modify their structure in function of pH, temperature and salt concentration [5]. For this reason, a variety of Pluronics is available on the market, differing for the molecular weight of the building blocks and for the hydrophobic–hydrophilic ratio, allowing the preparation of thermosensitive hydrogels with different properties, e.g., in terms of critical gelation concentration (CGC) and gelation time at physiological condition [3].

Thus, poloxamers represent a convenient choice in pharmaceutical technology and biomedical area due to their commercial availability, wide range of molecular weights, peculiar behavior and flexibility. Poloxamers are FDA approved and listed in the US and European Pharmacopoeia; they are non-toxic and non-irritant therefore they can be used as solubilizer, emulsifier, stabilizer, and administered through oral, parenteral, topical routes. As wetting agents, they are useful in ointments, suppository bases, and gels [6].

Poloxamer hydrogels use in drug release appeared at the beginning of the 1970s; they were a response in the search for new safer and faster treatments for the delivery of highly-effective therapeutic agents to a target cell, even if they have poor mechanical properties such as low tensile strength and low Young’s modulus that limit their practical applications, sometimes even as medical device coatings [7]. 

Analysis of recent literature covering a range of treatment pathways and diseases, reveals a major emphasis on “smart” drug carriers developed with poloxamers. The different range of potential delivery methods is highlighted in this review by discussing how the poloxamer solution behavior enables multiple formulation processing routes, drug-encapsulating structures, and engagement with physiological barriers to drug passage.

The first research work concerned the treatment of thermal burns [8] followed by researches on the release of hormones [9], tetracycline [10], proteins [11], and more recently for norcantharidin delivery [12], heparin [13], and anti-HIV drugs [14]. In the last twenty years the numbers of papers related to poloxamer hydrogels for biomedical applications is exponentially increased (Figure 1).

More recently such hydrogels have become especially attractive to the new field of “tissue engineering” as matrices for repairing and regenerating a wide variety of tissues and organs [15].

They have been successfully employed as scaffold-forming materials for cell printing technology, a computer-aided tissue engineering technology based on the layered deposition of cellularized hydrogels to form complex 3D constructs [16,17,18].

In this review the poloxamer hydrogels will be taken into consideration, giving an overview of their use as drug delivery systems (DDS) in different routes of administration, especially ophthalmic, transdermal, and vaginal ones. Finally, this paper focused on their potential in tissue and membrane regeneration, in the field of biomedical engineering and their application as micellar systems for gene delivery and cancer therapy. 

## 2. Poloxamers

### 2.1. Poloxamers Properties

These polymers are synthetic triblock copolymers of poly(ethylene oxide)-b-poly(propylene oxide)-b-poly(ethylene oxide) (PEO-PPO-PEO) (Figure 2). 

They are synthesized through the sequential addition of PO and EO monomers in the presence of an alkaline catalyst, such as sodium or potassium hydroxide, obtaining different copolymers with a different number of hydrophilic EO and hydrophobic PO units, which are also characterized by their distinct hydrophilic–lipophilic balance (HLB) value. Changes in the copolymer composition (PPO/PEO ratio) and molecular weight (PEO and PPO block length) during synthesis leads to macromolecular surface active agents with specific properties suitable in various technological areas.

Poloxamers appear in the form of liquids, pastes, and solids, depending on their molecular weights varying from 1100 to 14,000, offering a pool of more than 50 amphiphilic, water-soluble and polymorphic materials. Their water solubility is subjected by different ratio (1:9 to 8:2) between the chain of propylene oxide (PO) and ethylene oxide (EO) with hydrophobic and hydrophilic behavior, respectively [16]. These differences also alter the in vivo properties and interactions with cells and cell membranes, and provides high potential for the design of innovative nanomedicines and new biomaterials.

Poloxamers were the first block copolymers produced for industrial purposes, synthesized by Wyandotte Chemical Corporation in the late 1940s. Today they are commercially known by the trade names Pluronics^®^, Lutrol^®^, Kolliphor^®^ (BASF), Antarox^®^ (Rhodia), and Synperonics^®^ (Croda).

The original manufacturer BASF introduced a specific nomenclature for Pluronics consisting of a letter indicating the morphism of each copolymer: L (for liquid), P (for paste) or F (for flakes) followed by a number referring to the molecular weight of the PPO block (the first one or two) and the weight fraction of the PEO block (the last one).

For example, P123 and F127 have the same molecular weight of PPO (in the order of 4000) but P123 has 30% of PEO and F127 70% of PEO [17].

The physicochemical characteristics and applications of Poloxamers were firstly reviewed extensively by Alexandridis and Hatton in 1995 [16]. In Figure 3, a 3D poloxamer modified pluronic grid showing the distribution of different copolymers, gathered according their physical state, is reported.

The most significant physical properties of most common poloxamers, are reported in Table 1: average molecular weight, melting point expressed in °C, viscosity (Pa·s) measured at 25 °C, 60 °C, and 50 °C for pastes, liquids, and solids, respectively, surface tension at 0.1% at 25 °C (dyn cm^−1^), hydrophilic–lipophilic balance (HLB) [4,17,19,20].

Commonly used poloxamers include P188 (F-68 grade), P237 (F-87 grade), P338 (F-108 grade), and P407 (F-127 grade) types, which are freely soluble in water.

### 2.2. Poloxamers Behavior

The aqueous solution properties of poloxamers have been intensely studied and thoroughly reviewed owing to their unique behavior and benefit to myriad applications.

In water solutions, the amphiphilic character of copolymers lead the macromolecule to self-aggregate into micelles with an inner core constituted by hydrophobic blocks and an outer shell constituted by hydrophilic units. They are nano-sized structures, normally between 10 and 200 nm that appears at the critical micellization concentration (CMC) and at critical micellization temperature (CMT). CMC value of poloxamer aqueous solutions decreases with increasing temperature and number of PEO segments, indicating that polymers with a larger hydrophobic (PPO) domain form micelles at lower concentrations and temperatures [21].

Poloxamer water solutions exhibit temperature sensitivity, in particular a “thermoreversible gelation” [21] for sufficiently concentrated samples; they show a sol-gel transition around 37 °C (physiological temperature) and gel-sol transition around 50 °C, being able to produce thermoreversible gels, some already approved by the Food and Drug Administration, which present great interest in food additives, drug delivery carriers in cosmetics, pharmaceutical ingredients, and tissue engineering [5,17]

Several mechanisms have been proposed for this behavior, the first one was related to the gel transition that was due to changes in micellar properties (Figure 4). 

Thanks to their core–shell architecture, the hydrophobic core can act as a drug-loading site, creating a space for the encapsulation of hydrophobic drugs through the establishment of physical or chemical interactions. The properties of the outer shell and inner core have an influence in the drug release, that can promote an easier or sustained release of the drug. Due the properties above mentioned, polymeric micelles can transport several drugs, improving the circulation time, as well as the enhanced permeability and retention effect. Moreover, these systems exhibit low-risk of chronic toxicity since the polymeric micelles are disassembled in vivo, in single polymer chains that can be excreted by kidneys. 

In order to prepare polymeric micelles for drug encapsulation, different methods can be used, however, the more common are the direct dissolution, dialysis, evaporation or film method, freeze-drying, microphase separation, and the oil-in-water emulsion [22].

The method depends, mostly, on the solubility of the copolymer and the drug in an aqueous medium. Some other aspects are also important to an efficient drug incorporation, such as copolymer characteristics, molecular weight and HLB. The structural and chemical characterization of polymeric micelles are important features to take into account in the development of these nanocarriers, since they have a direct influence on the efficiency of these drug release forms, in terms of size, polydispersion index, zeta potential, encapsulation efficiency (EE), and drug loading capacity (DL) [23].

Other researchers [24] discussed gelation as a function of thermodynamical parameters. The enthalpy of gelation depends on CMC and temperature, the value change in the case of poloxamers gelation is unfavorable unlike the gelatine gelation where a great enthalpy change occurred.

The useful concentration for gel formation is the same as for poloxamers with the same PPO/PEO ratio and it decreases with the increasing of polymer molecular weight, denoting the significance of this parameter. While the presence of electrolytes reduces gel transition temperature, enthalpy of gel formation is not significantly changed by the addition of other substances. This behavior suggests that entropy plays the major role in the gelation process. 

For all these behaviors, the poloxamers were used for the study and for the development of innovative pharmaceutical forms in different administration routes.

## 3. Drug Delivery Systems (DDS)

Poloxamer sol-gel reversible hydrogels have attracted the attention for practical biomedical and pharmaceutical applications because of constituents solubility, biocompatibility with biological systems and easy administration of pharmaceutical formulations. The pharmaceutical and biomedical fields covered by the use of poloxamers including solubilization of hydrophobic drugs, controlled release, biomacromolecule delivery (e.g., proteins and genes) and tissue engineering.

Most applications involve the use of Poloxamer P407 and include delivery of protein/peptide drugs [25], such as insulin [26], interleukin-2 [27], epidermal growth factor [28], bone morphogenic protein [29], fibroblastic growth factor, and endothelial cell growth factor [30].

In recent years these hydrogels have been used as carriers for most routes of administration, the most interesting are discussed below.

### 3.1. Poloxamers for Oftalmic Administration

The thermoreversible gels have shown a growing interest at ocular level because they combine peculiar characteristics: i.e., the formulation when applied is sol (it performs like eye drops) and becomes gel with body temperature (it performs like ointment) increasing the in situ residence time. 

The most well-known ocular drug delivery system is eye drops, but they have a short residence time because they are quickly drained through the nasolacrimal route resulting in frequent dosage regimen which leads to an increase in side effects and poor patient compliance.

Poloxamers P407 and P188 are among the most commonly used in this case for their good water solubility, solution clearness, optimal viscosity, and ocular tissue safety.

Recent works have focused on the best formulation to obtain a hydrogel with useful features for ocular administration and without toxicity. About this point Al Khateb et al. [31] studied two formulations containing poloxamers, i.e., P407 and P188, with different gelation properties depending on concentration solutions and the ratio of their mixtures. Transparent gels were obtained only in the case of 20% w/w P407 and P188 solutions. Furthermore, these preparations were non-toxic or irritating to the corneal mucosa and then suitable for application in ocular drug delivery. 

Fathalla Z.M.A. et al. [32] studied the blend of the same two poloxamers for a controlled ocular delivery of ketorolac tromethamine (KT). The most promising gel formulations, loaded with KT, were those containing the mixtures of P407:P188 23:10 w/v% and 23:15 w/v% respectively. These gels do not present toxicity and do not irritate the conjunctiva and cornea.

In recent years other polymers have been added to poloxamers-based gels to obtain different drug release characteristics and to modify the rheological properties.

Among these works, Yu et al. [33] has to be remembered for the synthesis of a cross-linked hydrogel system containing carboxymethyl chitosan (CMC) and poloxamer P407, where the presence of CMC with biological properties could improve hydrogel biocompatibility. This formulation, containing nepafenac (NP) as a model drug, showed good rheological properties at gelation temperature (32–33 °C) and a sustained release of NP from hydrogel, so as to be considered a pH–temperature-responsive ophthalmic drug delivery system.

Another approach, with good results, involved the introduction of various colloidal carrier systems to easily load poorly soluble drugs into poloxamers hydrogels.

Lou et al. [34] incorporated curcumin-loaded albumin nanoparticles into a hydrogel based on a mixture of P407/P188 for local ocular administration, to treat diabetic retinopathy. This formulation, which became a gel when exposed to eye temperature, may be applied as eye drops. Nanoparticles provided the sustained drug release while the presence of hydrogel prolonged the in situ residence time.

Finally, Almeda et al. [35] reported the combination of lipid nanoparticles and a thermoresponsive polymer with mucomimetic properties (poloxamer P407). The incorporated nanoparticles showed an average size below 200 nm, a good positive zeta potential and an efficiency of ibuprofen encapsulation of about 90%. The optimal poloxamer concentration in thermoreversible gel was 15% (w/w) Pluronic^®^ F-127. The formulation did not present a relevant cytotoxicity and showed a sustained release of ibuprofen over several hours. The strategy proposed in this work can be successfully applied to increase bioavailability and therapeutic efficacy of conventional eyedrops.

### 3.2. Poloxamers for Transdermal Administration

Transdermal drug delivery is a valid alternative to the oral and parenteral route because it offers several advantages: Avoidance the hepatic first pass, good compliance by the patient, and easy access. The most studied formulations for this route contained poloxamer P407, as a polymer, and as drugs those with anti-inflammatory [36,37], analgesic [38], local anesthetic [39], and cardiac [40,41] activity, rarely are present preparations containing big molecules such as arginine, vasopressin, and insulin [42,43]. 

Generally in these topical preparations it is necessary to introduce an enhancer substance which is able to facilitate the passage through the stratum corneum (thickness of 10 to 15 μm) which is the main barrier to drug penetration.

In the last decade the penetration enhancers have been replaced by the microneedles (MNs) that have the capacity to permeate the stratum corneum and infuse the active ingredient in the deep areas of the skin.

Microneedles are needles similar to the ones useful for hypodermic injections but they present different sizes: from 1–100 microns in length and 1 micron in diameter (Figure 5). They are manufactured with silicon [44,45], metals such as stainless steel, palladium, nickel and titanium [46,47,48], carbohydrates including galactose, maltose, and polysaccharide [49,50,51], glass [52], ceramics [53], and various other polymers [54,55]. MNs are fabricated in backing that can be applied to the substrate like a patch carrying different drugs by penetrating through the skin, mucosal tissue and sclera [56,57,58,59]. Their dissolution has to be taken into consideration because it can influence drug delivery [60,61].

One of the most recent researches regards sol-gel transition property of poloxamers to obtain in situ forming hydrogel microneedles, for the delivery of methotrexate to treat solid tumors [62]. The use of this drug by transdermal route is generally limited by its relatively high molecular weight and hydrophilicity. For this purpose four formulations were prepared with two different methotrexate concentrations (0.2% and 0.4% w/w) with poloxamer P407 (20% w/w) and without, using the last one as a control, replacing the polymer with deionized water. In this study, it was confirmed the sol-gel transition of the formulations at skin temperature (32 °C), maintaining skin barrier function and skin viscoelastic properties after administration of the formulations. In vitro drug diffusion studies, using a Franz cell, showed that the formulations containing methotrexate (0.2% and 0.4%) without P407 released overall drug after 22 and 35 h respectively, while the one with P407 after 72 h. For this reason it is possible to conclude that the poloxamer-based formulations provided a steady and sustained delivery.

In an even more recent work, the formation of depots of thermoreversible poloxamers in skin micropores using MNs to transdermal drug delivery has been reported for the first time [63]. Sodium fluorescein (FS) was used as model drug to study in vitro permeation at different concentrations. In order to crate pores into the skin and to overcome the stratum corneum MNs have been used, then the drug loaded poloxamer solution was applied to fill pores, subsequently an in situ gelation at skin temperature of 32 °C occurred.

For this goal poloxamers P407, P237, and P338 were used at different concentrations (from 15 to 30% w/w in water) loading different amounts of fluorescein.

The formulations were characterized for their rheological properties and in situ gel formation. The distribution of FS in skin tissue was tracked by confocal laser microscopic analysis with higher intensity of FS in MN-treated skin tissues. The in vitro fluorescein release studies were carried out using vertical Franz diffusion cells. The release profiles indicated that the concentration of fluorescein (0.1%, 0.3%, and 0.5% w/w) was a variable parameter that significantly affects drug release as well as the type of poloxamer used. In particular P338 and P237 0.1–0.3% FS-loaded formulations provided a total drug release during 16 h while 0.5% FS-loaded provided a release for 20 h. Moreover, for a longer time (about 24 h), the release from P407 formulations was comparable. All the poloxamers depots started dissolving according to the dominant hydrophilic interactions and gels did not remain intact. It was concluded that P407 provided the best release for a longer duration and it was selected as the best drug delivery for in vitro permeation assays. These studies confirmed that drug loading is not a limitation and permeation of FS after MN treatment was found in more controlled manner and for a long time when compared to a permeation across untreated skin sample.

### 3.3. Poloxamers for Vaginal Administration

Another interesting pathway exploited in drug delivery systems is the vaginal route, allowing both the systemic and local absorption of drugs poorly absorbed after oral administration [65,66,67]. The vagina has a vast network of blood vessels that make easy the systemic absorption, avoiding deactivation at gastrointestinal level and hepatic first pass. Mucoadhesive pharmaceutical forms have been studied for this district due to the presence of mucus which increases permeability. The thermoreversible systems, appropriately modified with polymers that promote mucoadhesion, allow to obtain a sustained drug release and a good bioavailability without altering the vaginal physiology [68].

After application of poloxamers for vaginal administration occurring gelation favored a long permanence of drugs on the administration site to promote a drug controlled release [69]. Vaginal drug delivery is based on the exploitation of polymers which are able not only to gelify at physiological temperature but also to adhere to the vaginal mucosa improving the in situ residence time.

A brief description of recent reports in the literature, based on poloxamer formulations by vaginal route, is given below.

In the first work [70] the authors investigated a novel amphotericin B (AmB) release system in the form of nanosuspension loaded into a poloxamer P407/P188 hydrogel. P407 (20% w/w) and P188 (5% w/w) were dissolved in AmB nanosuspension, the AmB NPs thermogel were characterized regarding nanoparticle features (particle size, zeta potential, morphology) and gel behavior (rheology, stability, in vitro drug release, and in vitro and in vivo anti-Candida efficiency). The nanosuspension-gel combination has been necessary because nanoparticles alone tended to aggregate while they were more stable in the poloxamer hydrogel. When compared with other biodegradable thermosensitive hydrogel, such as polyester-based gels, poloxamer were not capable of an in vivo degradation. This property is really suitable for vaginal delivery because it means grater safety in use. Another very important element in the development of vaginal formulations was mucoadhesion; poloxamers have lower adhesion properties compared to compounds such as Carbopol, the addition of some bioadhesive materials into P407/P188 thermogel would be a feasible path [71,72,73]. Finally, the in vivo anti-Candida assay showed a better antifungal efficiency of AmB in the thermoreversible gel when compared with commercial effervescent tablets at the same drug dose (2.5 mg/Kg). 

A second paper [74] described a novel vaginal delivery strategy consisting of two pharmaceutical forms placed together to give an expansible thermal gelling aerosol foam (ETGFA) that combined the advantages of foam and gel penetration and carrier retention in vaginal canal, respectively.

ETGA was prepared adding an optimized amount of P407 (18–22% w/w) and P188 (0–5% w/w), achieved by evaluating the gelation temperature, the adhesive agents (arabic gum, sodium carboxymethyl cellulose, sodium alginate, and xanthan gum) and silver nanoparticles to obtain a drug concentration of 1% w/w. To study a better performance in foam expansion and duration, propane/butane 80/20 v/v and dimethyl ether were compared as propellants. The formulation was characterized in regards of rheology, foam expansion, adhesiveness and drug release. ETGA showed a better extended drug release (over 4 h) dose –dependent antimicrobial effects on the vaginal flora and no tissue irritation when compared to a commercial antimicrobial gel. These results indicated that ETGA could be a suitable formulation for vaginal drug delivery.

## 4. Tissue Regeneration Scaffolders

Tissue engineering, in this last decade, has emerged in the biomedical field because it allows researchers to create specific devices that represent in vivo tissues that can be replaced or increased to address current therapeutic challenges [75].

Poloxamers have received special attention for tissue regeneration based on their biocompatibility, low cytotoxicity, and good rheological properties [76]. 

In particular, the area the most explored deals with regeneration of bone tissue. The use of growth factors such as bone morphogenetic protein-2 (BMP-2) for bone repair has been reported [77,78] but it is very expensive and easily loose its integrity. Recently, other compounds that have received interest were statins because they altered bone metabolism through different mechanisms [79,80,81]. Their bioavailability after oral administration was low and a real bone healing cannot be expected. For this reason different delivery systems for local delivery of statins have been evaluated. The study of incorporation of rosuvastatin (RSV)-loaded chitosan/chondroitin sulfate nanoparticles into a thermosensitive hydrogel is reported below [82]. At first this research considered nanoparticles preparation and optimization, thereafter it dealt with the characterization of thermosensible hydrogels.

Gel formulation consisted of poloxamer P407 (18–20% w/v), hyaluronic acid (HA) 1–3% w/v and hydroxypropyl methylcellulose (2% w/v) to stabilize the formulation itself.

HA [83] and P407 had also positive effects on articular cartilage, simulating the regenerative process within the joint. The study showed that gel provided a low viscosity at 4 °C and gelification at 35 °C; drug release from nanoparticles, inserted into the hydrogel, was around 60% after 48 h, while it was completely released from nanoparticles alone within 12 h. This behavior indicated that the drug release was controlled and sustained.

Moreover, the hydrogel formulation showed an improvement in osteoblast viability and proliferation due to the used polymer and the biological properties of the nanoparticle delivery system.

Another paper [84] reported the design of a cryogel scaffold for the regeneration of an intervertebral disc tissue (nucleus pulposus NP). NP is located inside the vertebral discs and has the function of absorbing the pressure exerted on the spine and keeping the vertebrae separated. The symptoms of NP degeneration are pain and limited mobility of the extremities. It is composed of up to 90% water, type II collagen and proteoglycans [85]. This study focused on the preparation of a novel gelatin-P407 cryogel (in different ratio: 1:1,2:1, 4:1,5:1, 7:1, or 10:1 respectively) as an alternative to the spinal fusion procedure. The composite hydrogel was tested by the following assays: Pore analysis, swelling potential, stability, mechanical integrity and cellular infiltration. The inclusion of P407 in the cryogel was designed to increase swelling ratio, due its hydrophilic nature, since in the NP degradation there is a rapid water loss. All sample presented a high swelling ratio after 24 h, mechanical durability and stability for 28 days in a body-like environment. The 7:1 and 10:1 gel scaffolds showed the most ideal pore diameters and a profuse cell infiltration after only 14 days. 

In recent years, 3D printing technology has become important in the biomedical field and in particular the use of thermoreversible gels that with their sol-gel characteristics can be very useful in tissue regeneration [86]. 

The principle on which the 3D printer is based is to lay a filament of polymeric material which is deposited layer upon layer until the desired system is obtained in 3 dimensions.

Bioprinting systems can be classified into three types: Laser based, jetting based, and extrusion based; recent techniques such as magnetic bioprinting and electrohydrodynamic jetting have been used in tissue engineering.

Hydrogels requirements for 3D bioprinting of ideal engineered tissues should be the following [87]: High porosity, rapid gelation, shape retention, and immunological issue avoidance.

Thermoreversible hydrogels have been successfully applied thanks to their gelation characteristics as they quickly pass from sol-gel state and are also easily extruded from 3D printers [88].

P407 showed good printability [86] but it was not suitable for long-term cell viability. As reported by the recent work [89] it is used, combined with gelatin to create a biocompatible hydrogel for vascular channels, for molds, exploiting its excellent rheological behavior, under shear stress, and its elasticity.

In addition to bioprintability, biological properties of hydrogels played a crucial role in successful tissues regeneration [90]. One of the major disadvantages of available hydrogel materials was that they do not facilitate differentiation of cells into multiple linkages. Despite this, thermoresponsive hydrogels may be very useful for tissue engineering applications and they will become part of the advancement in bio printing technology.

## 5. Poloxamer as Micellar Systems

Polymeric micelles generated by poloxamers with peculiar characteristics and under specific conditions are exploited as nanometric drug carriers [91]. They have taken hold in recent years as they are able to solubilize poorly water-soluble drugs [36] and decrease undesirable cellular interactions [92,93]. In this review chapter, two recent applications (gene delivery and cancer therapy) of poloxamer micelles have been taken into account.

The development of gene delivery carriers has emerged as a promising technology in transporting genes directly to the target site as therapeutic factors [94]. Current gene transfer vectors used in regenerative medicine approaches included no viral [95] and viral vehicles [96]. The complexation of DNA with cationic polymers (polyplexes) or lipids (lipoplexes) protected DNA against degradation by nucleases and serum components creating a less negative surface charge. They can be designed to target specific cell types through receptor–ligand interactions [97], but these systems exhibited aggregation tendency, a low transfection efficiency and short time transgene expression levels [98]. The use of poloxamers has been described as a tool to increase efficiency of viral gene transfer, obtaining a localized delivery into targets or protecting the vectors.

In the last decade many works that focused the attention on gene delivery were published; the most recent and significant studies are here summarized.

Many researchers have studied the recombinant adeno-associated viral (rAAV) carriers as adapted gene transfer vectors to direct human cartilagine in regenerative medicine [99,100,101]. However, their use is limited cause their rapid neutralization by the presence of antibodies or heparin which prevent their binding to the viral receptor, present on the cell surface. In this regard, based on the use of polymeric biocompatible materials the combining of direct rAAV-mediated gene transfer with tissue engineering approaches, may offer efficient alternatives. 

In this study [102] the possibility of providing rAAV to cartilage regenerative cells via self-assembled poloxamers was evaluated. The formulations consisted of P188 and P407 (2% w/v i.e., above CMC)/rAAV or poloxamine/rAAV were directly incubated within monolayer cultures of hMSCs, cartilagine regenerative cells. At thisoptimized concentration the two poloxamers were more efficient in increasing gene expression over time than a treatment lacking them.

The highest rAAV gene transfer occurred when the PEO/PPO ratio was shifted to hydrophilicity (HLB > 24) using a concentration above CMC (2%) (89.5–94.6% efficiencies with up to 2.7-fold increase in transgene expression for at least 21 days, suggesting that micelles may be better carriers than unimers).

In another paper [103], lentiviral vectors (LV) used for the transduction of human and nonhuman cells were taken into consideration. The optimal transduction conditions for efficient LV gene delivery into target cells depended on a number of factors, including cell density, purity of lentiviral preparation, virus transduction units (TU), multiplicity of infection (MOI) and presence of adjuvants that facilitate transduction [104,105,106]. These researches identified and validated novel adjuvants for improving transduction efficiency, in particular five representatives poloxamers (P402, P235, P188, P407, and P338). P338 proved to be the best choice since it produced low toxicity and effective viral transduction, especially in difficult-to-transduce cells of T-cell origin.

Further application strategies with poloxamer regard cancer therapy [4]. Poloxamers have gained interest in the field of oncology because of their ability to developed stable systems with the capability of efficient drugs encapsulation and deliver. They usually present small sizes and self-assembling behavior in micellar systems in an aqueous medium, good stability in physiological medium and they avoid the deactivation by RES organs improving consequently drug bioavailability [22,107].

In this biomedical and pharmaceutical area the most recent works in the literature are reported. The first one concerned the use of poloxamer micelles containing a dye (CyFaP) as photo-acoustic imaging (PAI) contrast for mammary neoplastic tissue [108]. In this formulation, dye was inserted in P407 (10% w/v) micelle dropwise under constant stirring. Serum stability studies, cell viability assays, deep tissue penetration and in vivo biodistribution studies were performed. This work has shown that poloxamer micelles are effective for PAI as they penetrate more deeply into breast cancer tissue and are also useful for lymphatic mapping. 

The second work [109] explored the anticancer activity of Salinomycin (SAL), an antibiotic isolated from Streptomyces albus [110], that has recently been reported as being effective against tumor cells and mainly to inhibit growth of a specific cancer cell sub-population: CSC. The main goal of the research was the encapsulation of SAL into an innovative poloxamer micelles (PM) system and a further evaluation of in vitro PM-SAL delivery in bacterial and in eukaryotic tumor cells. PM formulations containing P407 and SAL in different amounts were defined with a two multivariate experimental design (DoE) to get the best preparation conditions. Micelles were characterized for size by dynamic light scattering, for their zeta potential and for the encapsulation efficiency. Their in vitro activity was conducted in bacterial cell culture and A549 (adenocarcinoma cell line) cell culture. In conclusion, these micelles seem to be able to deliver SAL into human cancer cells promoting a high drug intracellular accumulation, greatly decreasing toxicity on healthy cells.

Finally, a combined study between radiation therapy and systemic chemotherapeutics to provide both local radiosensitization and systemic control of tumor disease has been reported [111,112]. This study showed the development of a single micelles formulation consisting of multiple poloxamers (MPMs) containing PARP (poly ADP- ribose polymerase) inhibitor talazoparib (Tal) and PI3K (phosphoinositol-3-kinase) inhibitor buparlisib (Bup). When administered during a radiotherapy cycle, that would increase the therapeutic activity of the two drugs in a preclinical model [113]. The performed formulation was made of poloxamer mixture (P103 and P407) obtaining micelles (MPMs) via a modified nanoprecipitation method [114]. MPMs were characterized by dynamic light scattering, to determine size, zeta potential and polydispersity, and by transmission electron microscopy (TEM). 

It was also observed that MPMs modulate drug release depending on pH; this behavior is a very useful tool, as the change in pH at tumor level (pH 6.8) would result in a faster release than what occurs when they are circulating in the blood stream (pH 7.4). Furthermore, metabolic assays confirmed that the use of both Tal and Bup synergistically enhanced in vitro cytotoxicity. The in vitro therapeutic activity of the two drugs was studied by a colony-forming assay and a DNA damage assay. 

In vivo experiments have shown that the administration of this new MPMs formulation during the course of radiotherapy gave promising results into the enhanced tumor control. The results of this study confirmed that it is important to carry on the development and advancement of combination therapies, especially with the aim of enhancing the outcomes of patients in late-stage and metastatic disease.

## 6. Summary

Poloxamers are peculiar synthetic tri-block copolymers composed of two poly(ethylene oxide) units and a poly(propylene oxide) one with amphiphilic properties. The dual character, confers to the copolymers interesting properties that can be controlled and changed by different PPO/PEO ratios. These polymers exhibit thermogelling behavior at strategical physiological temperatures. 

The most used and studied poloxamer P407, at a concentration of 20% w/w in water, is able to convert the solution to a clear gel by warming the system from room (25 °C) to body temperature (near 37 °C). This thermoresponsive feature has made it attractive for several applications including injectable and topical pharmaceutical formulations, where the material may flow through an applicator or syringe before forming a gel upon contact with the body (35–37 °C). 

Because of their biocompatibility and nontoxicity poloxamers have been widely utilized in biomedical applications. In this review their usefulness as thermoreversible hydrogels in the most significant routes of administration (i.e., ocular, transdermic and vaginal route) was reported. Moreover, the latest frontiers have been highlighted for what concerns poloxamers employed in tissue engineering and in gene and cancer therapy.

## Figures and Tables

**Figure 1 pharmaceutics-11-00671-f001:**
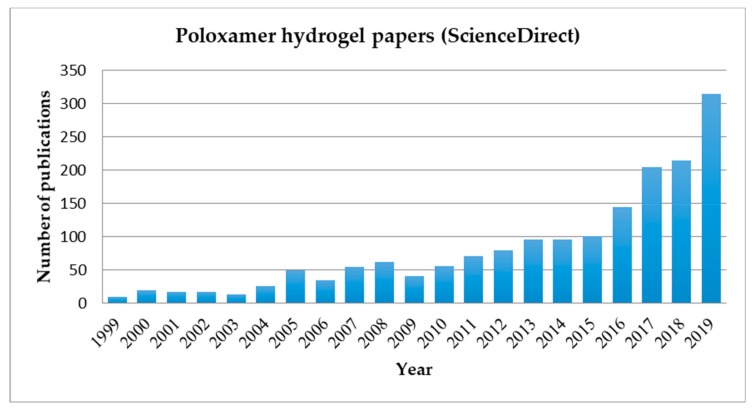
Histogram showing the increase in publications related to the keywords “poloxamer hydrogel” in Science Direct database during the past twenty years.

**Figure 2 pharmaceutics-11-00671-f002:**
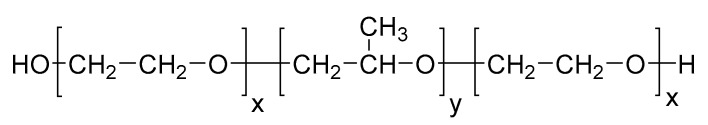
Chemical formula for poloxamers: x and y are the lengths of PEO and PPO: poly(ethylene oxide) and poly(propylene oxide) chains, respectively.

**Figure 3 pharmaceutics-11-00671-f003:**
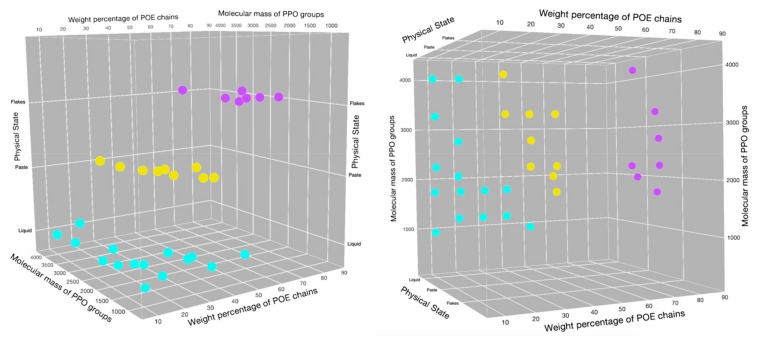
Poloxamers 3D distribution according to physical state (solid flakes = magenta; paste = yellow; liquid = blue), weight percentage of POE chains and molecular mass of the PPO groups (adapted from [16]).

**Figure 4 pharmaceutics-11-00671-f004:**
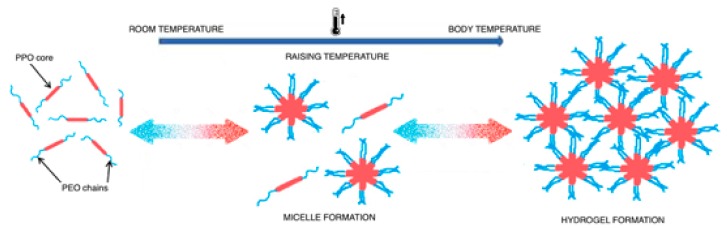
Schematic representation for hydrogel formation.

**Figure 5 pharmaceutics-11-00671-f005:**
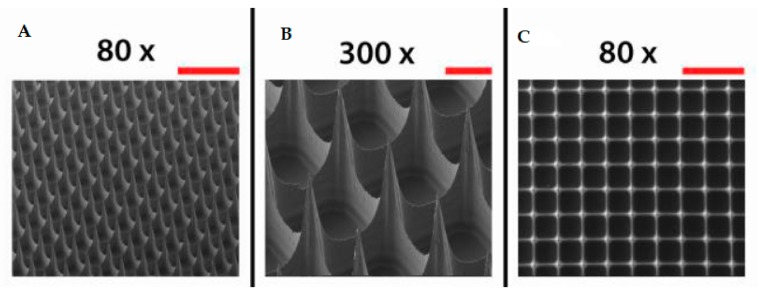
Scanning electron microscopy images of microneedle arrays imaged from a lateral view (**A**,**B**) and from the top side (**C**). Scale bars represent 500 µm (80×) (**A**,**C**) and 100 µm (300×) (**B**) (adapted from [64]).

**Table 1 pharmaceutics-11-00671-t001:** Properties of the most common poloxamer poly(ethylene oxide)-b-poly(propylene oxide)-b-poly(ethylene oxide) (PEO-PPO-PEO) copolymers.

Poloxamer	Pluronic	PEO%	Average Molecular Weight	Melting Point (°C)	Viscosity (Pa·s)	Surface Tension (dyn cm^−1^)	HLB
P105	L35	50	1900	7	0.375	49	18–23
P108	F38	80	4700	48	0.260	52	>24
P122	L42	20	1630	−26	0.280	46	7–12
P123	L43	30	1850	−1	0.310	47	7–12
P124	L44	40	2200	16	0.440	45	12–18
P182	L62	20	2500	−4	0.450	43	1–7
P183	L63	30	2650	10	0.490	43	7–12
P184	L64	40	2900	16	0.850	43	12–18
P185	P65	50	3400	27	0.180	46	12–18
P188	F68	80	8400	52	1.000	50	>24
P212	L72	20	2750	−7	0.510	39	1–7
P215	P75	50	4150	27	0.250	43	12–18
P217	F77	70	6600	48	0.480	47	>24
P234	P84	40	4200	34	0.280	42	12–18
P235	P85	50	4600	34	0.310	42	12–18
P237	F87	70	7700	49	0.700	44	>24
P238	F88	80	11,400	54	2.300	48	>24
P288	F98	80	13,000	58	2.700	43	>24
P333	P103	30	4950	30	0.285	34	7–12
P334	P104	40	5900	32	0.390	33	12–18
P335	P105	50	6500	35	0.750	39	12–18
P338	F108	80	14,600	57	2.800	41	>24
P402	L122	20	5000	20	1.750	33	1–7
P403	P123	30	5750	31	0.350	34	7–12
P407	F127	70	12600	56	3.100	41	18–23

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
