# Peer review of "Poloxamer Hydrogels for Biomedical Applications"

_pharmaceutics, 2019, doi:10.3390/pharmaceutics11120671_

Round 1

Reviewer 1 Report

It is useful to publish a review article on the biomedical applications of Pluronics.  However, this article is missing key information.  If Pluronics are to be used as an in vivo gelling material, their critical gelation temperature must be slightly below body temperature.   There are many different Pluronics surfactants available.  The critical gelation temperature depends upon Pluronics molecular weight, Pluronics concenetration, and Pluronics composition.  Hence for a given Pluronics molecule, how does one adjust the drug formulation to obtain a critical gelling temperature slightly below body temperature?

Author Response

Response to Reviewer 1

1) It is useful to publish a review article on the biomedical applications of Pluronics.  However, this article is missing key information.  If Pluronics are to be used as an in vivo gelling material, their critical gelation temperature must be slightly below body temperature.   There are many different Pluronics surfactants available.  The critical gelation temperature depends upon Pluronics molecular weight, Pluronics concenetration, and Pluronics composition.  Hence for a given Pluronics molecule, how does one adjust the drug formulation to obtain a critical gelling temperature slightly below body temperature?

Response 1:

We insert in the text the explanation of the related question (section 2): Thanks to the variety of thermoreversible poloxamers, it is possible to choose the most suitable mixture in order to obtain a gelling temperaturte in a range from 32 to 37 °C.

Reviewer 2 Report

The review paper describes poloxamer copolymers which have been used widely for biomedical applications. The authors focus is mainly the application of poloxamer hydrogels for design of drug delivery systems including micelles as well as fabrication of tissue engineered scaffolds. In general, this review paper is a nice study, easy to understand and may attract readers' attention after major revision. My suggestions to further enhance the manuscript are:

1- The introduction should specifically focus on thermoresponsive hydrogels (TH) to narrow the logic behind the paper. This may include a brief history of THs, their importance in biomedical, pharmaceutical and bioengineering applications as well as their major drawback and strength. The introduction in its present format is too wide to communicate.

2- It is necessary to inform potential readers about the methodology of the study including data bases that were used, selected keywords and the associated time frame (years) the were selected. The summary of this section may represent a brief statistical analysis about the number of publications per year/ selected keywords/ etc.

3- The review paper lacks the aim of the study as well as the novelty. Compared to other similar review studies/ publications, what is the message to take home

4- In section 2, A brief discussion to compare the advantages and disadvantages of poloxamers to some well-known thermoresponisve hydrogels (i.e. poly acryl amide) may highlight the importance of the study.

5- I would re-organize section 2, in order to explain the mechanism of formation of poloxamers, over a different range of viscosities, separately. This is an important part and may include schematics, figures or flowchart to further enhance the paper.

6- Since the paper describes the application of poloxamers in a wide range of biomedical engineering field including drug delivery systems and tissue engineering, the addition of table(s) to identify other properties (physico-chemical, mechanical, biocompatibility, etc.) may further enhance the study.

7- In general, the manuscript lacks relevant images and descriptive schematic drawings. I would make the paper more attractive to the readers with including proper images, graphs or schematic drawings. For example, SEM image (s) revealing the shape and size of the microneedles is suitable (lines 181-183).

Other minor concerns are

1- The paper needs significant English revision. Some examples are: lines 31-32, line 43 (should read cell encapsulation), line 44 (full stop at the end of sentence), lines 60-63, line 114, lines 115-116, line 132, line 258 (should read scaffolds), line 265. I mentioned some examples here and the remaining are the authors responsibility.

2- Please re-organise paragraphs, throughout the manuscript. Perhaps merging some of the short ones makes the manuscript more fluent and consistent.

3- Table 1 needs references.

4- Please check Ref 12.

Author Response

Response to Reviewer 2

The introduction should specifically focus on thermoresponsive hydrogels (TH) to narrow the logic behind the paper. This may include a brief history of THs, their importance in biomedical, pharmaceutical and bioengineering applications as well as their major drawback and strength. The introduction in its present format is too wide to communicate.

Response 1: According to referee suggestions, we deeply change Introduction summarizing general informations and focusing on THs. In the text we included a short THs history, major drawback and strength, highlighting the potential in biomedical application.

It is necessary to inform potential readers about the methodology of the study including data bases that were used, selected keywords and the associated time frame (years) the were selected. The summary of this section may represent a brief statistical analysis about the number of publications per year/ selected keywords/ etc.

Response 2: In order to better explain the value of the review methodology applied, we introduced Figure 1, an histogram showing the increase in publications related to the keywords “poloxamer hydrogel” in ScienceDirect during the past 20 years.

The review paper lacks the aim of the study as well as the novelty. Compared to other similar review studies/ publications, what is the message to take home

Response 3: The changes in the introduction section highlight better the aim and novelty of the review.

In section 2, A brief discussion to compare the advantages and disadvantages of poloxamers to some well-known thermoresponisve hydrogels (i.e. poly acryl amide) may highlight the importance of the study. I would re-organize section 2, in order to explain the mechanism of formation of poloxamers, over a different range of viscosities, separately. This is an important part and may include schematics, figures or flowchart to further enhance the paper.

Response 4-5: According to referee suggestions we re-organize section 2, explaining poloxamer synthesis, properties and behavior. In the section we added several advantages over other kind of thermoresponsive hydrogels, and we introduced an original poloxamer grid and a figure showing micellar  properties of pluronics.

Since the paper describes the application of poloxamers in a wide range of biomedical engineering field including drug delivery systems and tissue engineering, the addition of table(s) to identify other properties (physico-chemical, mechanical, biocompatibility, etc.) may further enhance the study.

Response 6: In the introduction and in Table 1 several useful properties are reported.

7- In general, the manuscript lacks relevant images and descriptive schematic drawings. I would make the paper more attractive to the readers with including proper images, graphs or schematic drawings. For example, SEM image (s) revealing the shape and size of the microneedles is suitable (lines 181-183).

Response 7: We introduced a SEM figure related to microneedles showing to different  magnifications (figure 5).

Other minor concerns are

The paper needs significant English revision. Some examples are: lines 31-32, line 43 (should read cell encapsulation), line 44 (full stop at the end of sentence), lines 60-63, line 114, lines 115-116, line 132, line 258 (should read scaffolds), line 265. I mentioned some examples here and the remaining are the authors responsibility. Please re-organise paragraphs, throughout the manuscript. Perhaps merging some of the short ones makes the manuscript more fluent and consistent.

We revised English language and paper layout.

3- Table 1 needs references. OK

4- Please check Ref 12. OK

Round 2

Reviewer 2 Report

The manuscript is now significantly enhanced and I support its publication.